# RETRACTED: High PGC-1α Expression as a Poor Prognostic Indicator in Intracranial Glioma

**DOI:** 10.3390/biomedicines12050979

**Published:** 2024-04-29

**Authors:** Yu-Wen Cheng, Jia-Hau Lee, Chih-Hui Chang, Tzu-Ting Tseng, Chee-Yin Chai, Ann-Shung Lieu, Aij-Lie Kwan

**Affiliations:** 1Department of Neurosurgery, Kaohsiung Veterans General Hospital, Kaohsiung 807, Taiwan; murraycheng1015@gmail.com; 2Graduate Institute of Medicine, College of Medicine, Kaohsiung Medical University, Kaohsiung 807, Taiwan; cychai@kmu.edu.tw; 3National Institute of Cancer Research, National Health Research Institutes, Tainan 701, Taiwan; 120307@nhri.edu.tw; 4Division of Neurosurgery, Department of Surgery, Kaohsiung Medical University Hospital, Kaohsiung 807, Taiwan; chchang20@gmail.com (C.-H.C.); cawaii7992@gmail.com (T.-T.T.); 5Department of Pathology, Kaohsiung Medical University Hospital, Kaohsiung 807, Taiwan; 6Department of Pathology, School of Medicine, College of Medicine, Kaohsiung Medical University, Kaohsiung 807, Taiwan; 7Department of Surgery, School of Medicine, College of Medicine, Kaohsiung Medical University, Kaohsiung 807, Taiwan; 8Department of Neurosurgery, University of Virginia, Charlottesville, VA 23806, USA

**Keywords:** glioma, PGC-1α, prognosis

## Abstract

Gliomas are the most common primary brain tumors in adults. Despite multidisciplinary treatment approaches, the survival rates for patients with malignant glioma have only improved marginally, and few prognostic biomarkers have been identified. Peroxisome proliferator-activated receptor γ (PPARγ) coactivator-1α (PGC-1α) is a crucial regulator of cancer metabolism, playing a vital role in cancer cell adaptation to fluctuating energy demands. In this study, the clinicopathological roles of PGC-1α in gliomas were evaluated. Employing immunohistochemistry, cell culture, siRNA transfection, cell viability assays, western blot analyses, and in vitro and in vivo invasion and migration assays, we explored the functions of PGC-1α in glioma progression. High PGC-1α expression was significantly associated with an advanced pathological stage in patients with glioma and with poorer overall survival. The downregulation of PGC-1α inhibited glioma cell proliferation, invasion, and migration and altered the expression of oncogenic markers. These results conclusively demonstrated that PGC-1α plays a critical role in maintaining the malignant phenotype of glioma cells and indicated that targeting PGC-1α could be an effective strategy to curb glioma progression and improve patient survival outcomes.

## 1. Introduction

Gliomas are the most common primary brain tumors in adults [1] and have a poor prognosis [2,3,4]. Although multidisciplinary treatment strategies have been applied, the survival of patients with malignant glioma has improved only slightly due to the high invasive ability and resistance to conventional treatments [5,6]. Few valuable molecules have been validated and widely applied as prognostic indicators in clinical practice, and a better understanding of the biological and molecular factors involved in gliomas is necessary for biomarker development [7].

Metabolic reprogramming is considered a hallmark of cancer [8]. Glycolysis, mitochondrial respiration, glutaminolysis, and fatty acid metabolism are important processes in cancer development [9,10,11,12,13]; they provide cancer cells with an adaptable metabolic feature and improve cancer cell survival under stress [14,15,16]. Mitochondrial biogenesis is critically regulated by the peroxisome proliferator-activated receptor γ (PPARγ) coactivator (PGC) family, which consists of PPARγ coactivator-1 alpha (PGC-1α) (PPARGC1A), PGC-1β (PPARGC1B), and PRC (PPRC1) [17]. Among numerous regulators or mediators of cancer metabolism, PGC1α is emerging as an essential controller of multiple metabolic pathways [18,19]. PGC1α functions as a necessary adaptor for cells to maintain a metabolic balance under harsh situations. It has protective effects against chronic diseases, such as skeletal muscle atrophy, heart failure, neurodegeneration, obesity, diabetes, and hepatic steatosis, some of which are predisposing factors for cancer [18,19,20]. There is recent evidence that PGC1α expression is closely associated with cancer progression. The exceptional ability of PGC1α to manipulate cellular metabolism enables cancer cells to thrive under a constantly fluctuating energy status and highlights the importance of PGC1α in effective cancer therapy [21]. PGC1α promotes carcinogenesis in the chemical-induced colon and liver carcinoma mouse models [22]. The ectopic expression of PGC1α has been observed in several cancer types [23,24,25,26], and PGC1α is regulated by several oncogenes and signaling pathways [27,28,29]. Hence, the role of PGC1α in glioma has not been determined. This study aimed to validate the clinical and pathological roles of PGC1α in glioma.

## 2. Materials and Methods

### 2.1. Patients

Patients with glioma at the Neurosurgery Department of Chung Ho Hospital, Kaohsiung Medical University (KMUHIRB-E(I)-20190150) were included. Patients diagnosed based on biopsies alone or with incomplete medical records, no follow-up visits, low-quality pathological results, or poor immunohistochemical staining were excluded. A total of 68 patients were finally included in this study.

### 2.2. Immunohistochemistry Staining

For every case, 3 µm sections were cut from formalin-fixed, paraffin-embedded tissue blocks. These sections were deparaffinized, rehydrated, and autoclaved at 121 °C for 10 min in Target Retrieval solution, pH 6.0 (DAKO, Glostrup, Denmark; S2369) to retrieve antigens. After 20 min at room temperature, 3% hydrogen peroxide was applied to the section for 5 min to block endogenous peroxidase at room temperature. After washing twice with Tris buffer, the sections were incubated with a 1:200 dilution of PGC1α for 1 h at room temperature. After washing twice with Tris buffer, the sections were incubated with the secondary antibody conjugated with horseradish peroxidase for 30 min at room temperature. Finally, the slides were incubated in 3,3-diaminobenzidine (Dako, Glostrup, Denmark; K5007) for 5 min, followed by Mayer’s hematoxylin counterstaining for 90 s and mounting with Malinol. For the IHC scoring system, samples were classified into two types based on staining intensity: low expression and high expression. The proportion of positive tumor cells was scored as follows: 0 (no positive tumor cells), 1 (<10%), 2 (10–50%), and 3 (>50%). The intensity of staining was determined as 0 (no staining), 1 (weak staining), 2 (moderate staining), and 3 (strong staining). The staining index (SI) was calculated as the product of the intensity and percentage of positive tumor cells, resulting in scores of 0, 1, 2, 3, 4, 6, and 9. A cut-off value of 4 for the total score was established; scores of ≥4 indicated high PGC1α expression and scores of <3 indicated low PGC1α expression.

### 2.3. Cell Culture

Cell lines utilized in this study were sourced from the American Type Culture Collection Cell Line Bank and maintained under conditions of 5% CO_2_ at a temperature of 37 °C. The GBM8401, GBM8901, and DBTRG-05MG cell lines were propagated in RPMI medium enriched with 10% fetal bovine serum (FBS). U87-MG and SVGp12 cell lines were grown in minimum essential medium supplemented with 10% FBS. Additionally, the G5T/VGH, Hs683, and A172 cell lines were cultivated in Dulbecco’s modified Eagle’s medium (DMEM) with 10% FBS. The M059K cell line was cultured in a 1:1 mixture of DMEM and F12 medium with 10% FBS. The cell lines, including GBM8401, GBM8901, U87-MG, G5T/VGH, DBTRG-05MG, M059K, Hs683, and A172, were derived from tissues of patients with glioblastoma (GBM), while the SVG cell line, used as a normal control, originated from healthy brain tissue.

### 2.4. Transfection

PGC1α siRNA in glioma cells was achieved using DharmaFECT Transfection Reagent (Dharmacon, Lafayette, CO, USA). The sequence of human PGC1α siRNA (Sigma, St. Louis, MO, USA) included siRNA#1: GAGAAUUCAUGGAGCAAUA, siRNA#2: GAAGAGCGCCGUGUGAUUU, siRNA#3: ACACUCAGCUAAGUUAUAA, and siRNA#4: GCAGGUAACAUGUUCCCUA. For transfection, 5 μM PGC1α siRNA was used. Following transfection with siRNA, cells were cultured for 3 days before use. PGC1α protein expression was detected by western blot analyses.

### 2.5. Cell Viability

Cells were added to DMEM containing 10% serum and placed in a 24-well plate with approximately 1 × 10^4^ cells per 0.5 mL in each well. These cells were incubated under 5% CO_2_, saturated humidity, and 37 °C for 24 h. Cells were counted using an MTT assay after co-culture with 5 μM PGC1α siRNA for 24, 48, and 72 h.

### 2.6. Western Blotting

All samples were lysed in 200 μL lysis buffer. Then, 50 μg of protein for every sample was loaded for separation by SDS-PAGE at 50 V for 4 h. The protein was transferred from the gel to a PVDF membrane. After 1 h in blocking buffer, the membranes were incubated with primary antibodies [PGC1α (1:500; Biorbyt, Ely, UK; orb13647), β-actin (1:20,000; Sigma; A5441), N-cadherin (1:500; Proteintech, Rosemont, IL, USA; 22018-1-AP), E-cadherin (1:500; Proteintech, Rosemont, IL, USA; 20874-1-AP), cyclin D1 (1:500; Proteintech; 60186-1-lg), VEGF (1:500; ABGENT, San Diego, CA, USA; P15692)] for 2 h at room temperature and secondary antibodies [Goat anti-Rabbit (1:2000; Millipore, Billerica, MA, USA; AP132P) and Goat anti-Mouse (1:2000; Millipore; AP124P)] for 90 min. ECL solution (Western Lightning, Newton Abbot, UK; 205-14621) was used to detect specific bands with MINICHEMI (Thermo, Waltham, MA, USA).

### 2.7. Invasion Assay In Vitro

Cell invasion was evaluated using a Transwell assay (CORNING Inc., Corning, NY, USA; COR3452) in vitro. Cells were seeded at 1 × 10^4^ per insert, and the lower chamber of the Transwell was filled with 2 mL of medium with nonsense siRNA (si-non) or PGC1α siRNA. After incubation for 24 h, cells remaining on the upper surface of the Transwell membrane were removed using a cotton swab. Cells that had passed through the Transwell to the bottom of the insert were fixed, stained, photographed, and quantified by counting in six random high-powered fields.

### 2.8. Migration Assay In Vitro

Cell invasion was evaluated using a wound healing assay (ibidi; Gräfelfing; Germany; 80209). Briefly, 24-well plates were coated in Matrigel, and cells were cultured at 37 °C for 12 h. Then, 70 μL of cells at 1 × 10^6^/mL was incubated with siRNA for 24 h. Images were obtained at 0, 8, 12, and 24 h.

### 2.9. Animal Model

GBM8401 cells, engineered to express fluorescence (1 × 10^5^ cells in 5 μL), were intracranially implanted into the striatum of immunodeficient mice (LASCO Laboratory Animal Center, Taipei, Taiwan). The animal protocol received approval from the Kaohsiung Medical University Committee of Institutional Animal Research (IACUC 108285). The mice were kept under constant environmental conditions with a temperature maintained at 24 °C and subjected to regular light/dark cycles (12 h each), with unrestricted access to a standard diet. For this experiment, the control group received injections of unmodified GBM8401 cells (*n* = 12), and the experimental group was treated with GBM8401 cells in which PGC-1α had been knocked down using shRNA targeting PGC-1α (*n* = 12). The mice were anesthetized using isoflurane, and fluorescence detection was conducted using the Xenogen IVIS^®^ Spectrum Noninvasive Quantitative Molecular Imaging System (J&H; Shenzhen, China; IVIS Lumina LT 2D) at intervals of 7, 14, and 21 days following the cell injections.

### 2.10. Data Analysis

SPSS 26.0 (IBM, Armonk, NY, USA) was used for statistical analyses. Chi-square tests were performed to evaluate correlations between PGC1α protein expression and clinicopathological parameters. The survival rate was analyzed using the Kaplan–Meier method with the log-rank test. Multivariate Cox regression analyses were used to verify the independent effect of each variable. The western blotting results were analyzed using Lane 1D. A *p*-value of <0.05 was considered statistically significant.

## 3. Results

### 3.1. Associations of PGC-α Expression with Clinicopathological Characteristics in Patients with Intracranial Glioma

Ninety-five patients from Kaohsiung Medical University Hospital diagnosed with intracranial gliomas were included. Clinicopathological characteristics, such as age, sex, World Health Organization (WHO) pathological classification, tumor size, receipt of chemotherapy and radiotherapy, and Karnofsky performance score (KPS) (a quality-of-life index), were evaluated in relation to PGC-1α expression levels. Among patients under 60 years of age, 30.5% showed low PGC-1α expression, and 44.2% exhibited high expression. Among patients over 60 years of age, 7.4% had low PGC-1α expression, and 17.9% had high expression. The age-related difference in PGC-1α expression was not statistically significant (*p* = 0.342). Regarding sex, 21.1% of male patients had low PGC-1α expression, and 33.7% had high expression. For female patients, 16.8% had low, and 28.4% had high PGC-1α expression, with no significant difference in expression based on sex (*p* = 1.000). The WHO grade showed a significant correlation with PGC-1α expression. In particular, 15.8% of patients with grade II glioma had low expression, and 8.4% had high expression, while in grades III/IV gliomas, 22.1% had low expression, and 53.7% had high expression (*p* = 0.003). No significant differences were found in PGC-1α expression with respect to tumor size, radiotherapy, chemotherapy, or KPS score (Table 1). In summary, there was a significant association between PGC-1α expression and pathological staging in patients with glioma, where higher PGC-1α expression was correlated with higher tumor grades.

### 3.2. Prognostic Significance of PGC-1α Expression in Glioma

As shown in Figure 1, a Kaplan–Meier analysis and subsequent log-rank test confirmed that PGC-1α expression is correlated with survival in astrocytoma; in particular, a high level of PGC-1α expression was significantly correlated with poor overall survival. In univariate analyses, the impact of each factor on the prognosis of patients with glioma was assessed. Patient outcomes were not significantly correlated with age (relative risk = 0.648, 95% confidence interval (CI), 0.311–1.351, *p* = 0.247), sex (relative risk = 0.892, 95% CI, 0.473–1.682, *p* = 0.724), tumor size (relative risk = 1.114, 95% CI, 0.572–2.171, *p* = 0.751), chemotherapy (relative risk = 1.542, 95% CI, 0.776 to 3.065, *p* = 0.217), or KPS (relative risk = 1.167, 95% CI, 0.586 to 2.324, *p* = 0.660). The WHO grade was identified as a significant prognostic factor (relative risk = 0.382, 95% CI, 0.171–0.851, *p* = 0.019) (Table 2). Additionally, radiotherapy had a relative risk of 1.845 with a 95% CI of 0.965–3.526 (*p* = 0.064) (Table 2), indicating a trend towards significance, suggesting that it has an impact on outcomes.

PGC-1α expression was closely related to prognosis (relative risk = 0.177, 95% CI, 0.075–0.417, *p* < 0.001), highlighting its potential as a critical prognostic factor in glioma. In a multivariate analysis, PGC-1α expression remained significant (relative risk = 0.197, 95% CI, 0.080–0.482, *p* < 0.001). However, the WHO grade, while showing a reduced relative risk of 0.546 and a 95% CI of 0.223–1.335, was not a significant factor in this analysis (*p* = 0.185) (Table 2). Therefore, PGC-1α was identified as an independent biomarker for prognosis in glioma.

### 3.3. Elevated PGC-1α Protein Expression in GBM Cells Relative to Normal Cells

To elucidate the difference in PGC-1α expression between normal glial cells and GBM, real-time PCR was conducted (Figure 2). *PGC-1α* mRNA levels were evaluated across various glioma cell lines, including GBM8401, U87-MG, G5T/VGH, DBTRG-05MG, M059K, and A172, and compared with levels in normal primary human astrocytes (SVGp12). PGC-1α expression levels in the glioma cell lines GBM8401, U87-MG, G5T/VGH, DBTRG-05MG, M059K, and A172 were significantly higher than those in SVGp12, suggesting that PGC-1α in integral in the pathogenesis of glioma. In comparison with levels in SVGp12, the average increases in PGC-1α expression in various glioma cell lines were as follows: 5.4-fold in GBM8401 cells, 1.7- to 2-fold in U87-MG cells at different time points, and approximately 4.6-fold in A172 cells. G5T/VGH cells displayed a moderate 1.5-fold elevation, DBTRG-05MG cells exhibited an approximately 2.5-fold increase, and M059K cells had an average increase of about 3-fold over levels in SVGp12. Given the high PGC-1α protein levels in GBM8401 and A172 cells, they were chosen for further analyses of the impact of PGC-1α downregulation on glioma progression. In particular, real-time PCR and western blot analyses were utilized to quantify the impact of PGC-1α siRNA silencing (Figure 3). In GBM8401 cells, there was a notable reduction in *PGC-1α* mRNA expression to an average of approximately 0.23 (relative to levels in wild-type cells) after siRNA treatment. Western blotting further demonstrated a significant decrease in PGC-1α protein levels (i.e., 0.35) following siRNA application, indicating effective knockdown. For A172 cells, real-time PCR revealed a decrease in *PGC-1α* mRNA levels to an average of 0.38 with siRNA treatment. Western blotting results echoed this finding, with PGC-1α protein levels diminishing to an average of 0.20 in comparison with levels in the control. These findings underscore the efficiency of siRNA targeting PGC-1α in downregulating its expression at both the mRNA and protein levels.

### 3.4. PGC-1α Knockdown Inhibited GBM Cell Proliferation

To evaluate the effect of PGC-1α on GBM cell survival, we compared cell viability in GBM8401 and A172 cell lines with or without the siRNA-mediated knockdown of PGC-1α. Viability was measured at 24, 48, and 72 h after transfection. As shown in Figure 4, in GBM8401 cells, there was a discernible decrease in cell viability to an average of 58% at 48 h post-transfection with si-PGC-1α. This reduction became more pronounced at 72 h, with an average viability of about 27%. In the A172 cell line, viability also decreased after si-PGC-1α transfection, averaging 42% at 48 h and approximately 27% at 72 h. These results highlight the significant impact of PGC-1α knockdown on the viability of GBM cell lines, affirming the potential value of PGC-1α as a target for therapeutic strategies. These data indicate that suppression of PGC-1α can markedly hinder the survival of GBM cells.

### 3.5. PGC-1α Knockdown Inhibited GBM Cell Invasion and Migration

To analyze cell migration, a wound-healing assay was employed with GBM8401 and A172 cells to observe the effects of PGC-1α suppression (Figure 5). PGC-1α was silenced using siRNA, and the migration percentage was recorded at 0, 8, 12, and 24 h post-transfection. For GBM8401 cells, the average migration percentages in the control group were approximately 1.9% at 0 h, escalating to about 32% at 8 h, 44% at 12 h, and approximately 78% at 24 h. In the si-PGC-1α group, the cells exhibited a significant impairment in migration, with average percentages of approximately 1.8% at 0 h, 16% at 8 h, 28% at 12 h, and 46% at 24 h post-transfection. Similarly, A172 cells demonstrated an average initial migration percentage of approximately 1.4%, with increases to approximately 19% at 8 h, 33% at 12 h, and 49% at 24 h in the control group. Post-transfection with si-PGC-1α, these cells showed a reduction in migration to averages of approximately 1.4% at 0 h, 7% at 8 h, 17% at 12 h, and 29% at 24 h.

Cell invasion capabilities for GBM8401 and A172 cell lines were evaluated to determine the effect of PGC-1α silencing on GBM invasiveness using a Matrigel invasion assay (Figure 6). Following transfection with siRNAs targeting PGC-1α, both cell lines exhibited a significant decrease in invasion rates compared with those in the control and negative control groups. In the GBM8401 cell line, the average percentage of invasion for the control group was approximately 71%, while si-PGC-1α transfection led to a significant reduction, with an average invasion rate of approximately 35%. For the A172 cell line, the average invasion rate in the control group was approximately 51%, and si-PGC-1α treatment resulted in a marked decrease to an average of approximately 11%. These results suggest that PGC-1α contributes to the invasive capacity of GBM cells, with its knockdown by siRNA significantly inhibiting invasion. This indicates that PGC-1α could be a promising target for therapeutic strategies aimed at mitigating GBM invasiveness.

### 3.6. Modulation of Carcinogenic Marker Expression by PGC-1α Knockdown in Glioma Cells

The effects of PGC-1α on key molecular markers associated with glioma progression were assessed in GBM8401 and A172 cell lines using western blot analyses (Figure 7). PGC-1α silencing through siRNA led to observable changes in the expression levels of several critical proteins. In GBM8401 cells, siRNA-mediated knockdown of PGC-1α resulted in marked reductions in the expression of N-cadherin (0.10-fold), Cyclin D1 (0.20-fold), and VEGF (0.33-fold), while E-cadherin expression showed an increase (2.85-fold) over levels in the control and negative control groups. For A172 cells, following PGC-1α knockdown, similar trends were observed. There were significant decreases in the expression of N-cadherin, Cyclin D1, and VEGF, with average fold-changes of approximately 0.33, 0.17, and 0.26, respectively. E-cadherin expression increased with an average fold change of approximately 4.28. These patterns suggest that PGC-1α has a substantial role in the regulation of molecules that are pivotal for glioma cell invasiveness and proliferation, providing insight into how PGC-1α suppression could alter the behavior of glioma cells and serve as a therapeutic target.

### 3.7. Effects of PGC-1α Knockdown on Tumor Growth and Survival in a Murine Glioblastoma Model

The effect of PGC-1α knockdown on tumor dynamics was monitored in vivo using a fluorescence-based tumor growth assay with a murine model. Mice were implanted with GBM cells expressing normal levels of PGC-1α (control group) or with PGC-1α knockdown. Fluorescence intensity measurements at days 7, 14, and 21 post-implantation revealed that tumor growth was markedly lower in the PGC-1α knockdown group than in the control group. This significant reduction in fluorescence intensity in the PGC-1α knockdown group was correlated with decreases in tumor activity and progression, implying that PGC-1α is integral to tumor growth within this biological system (Figure 8). These data further suggest that PGC-1α is a therapeutic target for inhibiting tumor growth and extending survival in GBM.

## 4. Discussion

PGC-1α, a transcriptional coactivator that orchestrates mitochondrial biogenesis and metabolic flux, has emerged as a pivotal factor in the metabolic reprogramming of cancer cells—a key adaptation that supports their rapid growth and survival under diverse conditions [17,30]. In several types of carcinomas, including breast, colon, and ovarian cancers, PGC-1α is frequently downregulated, and its downregulation is associated with a loss of metabolic flexibility and an increased reliance on glycolysis for energy production, even in the presence of oxygen, a phenomenon known as the Warburg effect [31,32,33]. The decreased expression in these contexts suggests that of PGC-1α is involved in maintaining metabolic checks that prevent cancer cell proliferation, thereby underscoring the importance of its tumor-suppressive functions [34]. Conversely, the upregulation of PGC-1α in certain cancers illustrates the adaptability of cancer cells in exploiting physiological mechanisms for their survival and expansion, contributing to enhanced mitochondrial biogenesis and metabolic efficiency, which provide cancer cells with the energy and biosynthetic precursors necessary for rapid growth [35,36]. Cancer cells undergo a fundamental shift in their metabolism to support increased demands for energy and biosynthetic materials, with PGC-1α standing at the crossroads of this metabolic reprogramming, regulating key pathways involved in energy production and the synthesis of macromolecules [37]. The roles of PGC-1α in mitochondrial biogenesis and function are a double-edged sword in the context of cancer [38]. On the one hand, the ability of PGC-1α to enhance mitochondrial efficiency can support the energetic and biosynthetic needs of rapidly proliferating cancer cells [35]. On the other hand, the role of PGC-1α in oxidative metabolism can lead to the increased production of reactive oxygen species (ROS), which can damage cancer cells and potentially limit their growth [30,39]. The Warburg effect, characterized by increased glycolysis in the presence of oxygen, is a hallmark of many cancers [40]. While PGC-1α is traditionally associated with oxidative metabolism, its influence on cancer metabolism extends to glycolytic pathways as well [33,41]. Through its regulatory network, PGC-1α can impact the expression of enzymes involved in glycolysis, thereby affecting the metabolic strategy of cancer cells [17]. PGC-1α emerges as a pivotal figure in cancer biology, influencing a spectrum of metabolic processes and playing contrasting roles in tumor progression and response to therapy. In melanoma, its expression level bifurcates cells into two distinct metabolic states, with high PGC-1α levels fostering oxidative metabolism and resistance to stress, whereas low levels render cells glycolytic, prone to apoptosis, yet more invasive [25,27,42,43,44]. Breast cancer cells leverage PGC-1α’s regulatory interactions to enhance mitochondrial function and drive growth, with implications for aggressiveness and resistance to treatment [24,35]. Co-activation between PGC-1α and c-MYC balance steers cell fate between survival and death, profoundly affecting treatment responses in pancreatic adenocarcinoma [29]. Similarly, in prostate cancer, PGC-1α dictates divergent paths of tumor behavior, with its levels modulating cell proliferation and invasiveness [23,45,46]. In our study, PGC-1α expression was significantly correlated with the pathological stage of gliomas, with higher expression in advanced tumors. While age, gender, tumor size, and treatments (e.g., chemotherapy and radiotherapy) did not significantly affect PGC-1α levels, the WHO grade was a notable exception, with support for its prognostic value. Kaplan–Meier and log-rank analyses further established PGC-1α as a crucial prognostic marker, with high expression levels associated with poorer overall survival in patients with glioma, suggesting that it is an independent biomarker for glioma prognosis.

PGC-1α is also closely related to oncogenic processes within astrocytic cells [47]. The elevation of PGC-1α and mitochondrial transcription factor A (TFAM) expression in astrocytoma tissues is a potential adaptive mechanism where cancer cells exploit mitochondrial biogenesis and enhanced energy production to support their rapid growth and survival in the hostile tumor microenvironment [48]. This scenario is further complicated by the positive correlation between PGC-1α levels and the activity of AMP-activated protein kinase (AMPK) and its phosphorylated form, suggesting a sophisticated regulatory network that promotes the invasive growth of astrocytomas [49]. In our result, knocking down PGC-1α in GBM cells led to significant reductions in their proliferation, migration, and invasion, as well as altered expression levels of key carcinogenic markers, pointing to the protein’s critical role in tumor growth and aggressiveness. In vivo experiments further demonstrated that PGC-1α knockdown drastically curbs tumor growth in a murine model, further indicating that PGC-1α is a viable target for GBM therapy. These findings collectively emphasize the importance of PGC-1α in maintaining the malignant phenotype of GBM cells and suggest that targeting PGC-1α could be an effective approach to inhibit GBM progression and improve patient outcomes. The limitation of our study is the insufficient exploration of the mitochondrial signaling pathways that are involved in the regulatory effects of PGC-1α on GBM progression. To address this limitation, we plan to incorporate specific inhibitors and employ bioenergetic assessment tools, such as the Seahorse Bioanalyzer, in our future work. This approach will enable us to precisely delineate the pathways through which PGC-1α influences tumor progression in GBM. By conducting these studies, we aim to deepen our understanding of the intricate mitochondrial mechanisms that are influenced by PGC-1α, thereby further validating its potential role as a therapeutic target.

## 5. Conclusions

PGC-1α plays a crucial role in GBM progression, influencing cell proliferation, invasion, migration, and the expression of oncogenic markers. The knockdown of PGC-1α in GBM cell lines, such as GBM8401 and A172, resulted in marked reductions in cell viability and migratory and invasive capabilities, as well as altered expression levels of key proteins associated with cancer progression. These in vitro findings were substantiated by in vivo evidence showing reduced tumor growth in a murine model with PGC-1α knockdown, highlighting the significance of the protein in tumor development and maintenance. Collectively, the results support the important role of PGC-1α in GBM and suggest that targeting PGC-1α could serve as an effective therapeutic strategy to impede GBM progression and improve outcomes. This research underscores the potential of PGC-1α not only as a prognostic biomarker but also as a promising target for the development of novel GBM treatments.

## Figures and Tables

**Figure 1 biomedicines-12-00979-f001:** Correlation Between PGC-1α Expression Levels and Patient Survival in GBM. (**A**) Immunohistochemical staining for PGC-1α expression in glioma. Representative immunohistochemical staining results for gliomas with high (left) and low (right) PGC-1α expression. (**B**) Kaplan–Meier survival analysis correlating PGC-1α expression levels with patient survival outcomes, assessed by the log-rank test. There was a significant difference in prognosis between the high and low-expression groups.

**Figure 2 biomedicines-12-00979-f002:** Relative *PGC-1α* mRNA Expression Between Glial and GBM Cell Lines. Comparison of *PGC-1α* mRNA expression between normal glial and GBM cell lines. The bar graph represents the relative expression levels of *PGC-1α* mRNA in normal glial cells (SVGp12) and various GBM cell lines (GBM8401, U87-MG, A172, G5T/VGH, DBTRG-05MG, and M059K), as measured by real-time PCR. Expression levels are normalized against the SVGp12 glial cell line, which served as the control. A statistically significant increase in expression was observed in the GBM8401 and A172 cell lines (* *p* < 0.05, ** *p* < 0.01) when compared with levels in SVGp12.

**Figure 3 biomedicines-12-00979-f003:** Effects of PGC-1α Knockdown on mRNA and Protein Expression in GBM Cells. (**A**) Relative *PGC-1α* mRNA levels in GBM8401 and A172 cells (*n* = 3). The control group was untreated cells. The negative control group was cells treated with non-targeting siRNA. The si-PGC-1α group included cells transfected with PGC-1α siRNA. mRNA levels were significantly lower following PGC-1α knockdown than the control group for both GBM8401 and A172 (*** *p* < 0.001, ** *p* < 0.01). (**B**) Western blot images and quantitative analysis of PGC-1α protein expression (*n* = 3). The protein bands for PGC-1α and β-actin are shown for GBM8401 and A172 cells. Quantification relative to β-actin demonstrates significantly lower levels of PGC-1α in the si-PGC-1α group than in the control group for both GBM8401 and A172 (*** *p* < 0.001, ** *p* < 0.01).

**Figure 4 biomedicines-12-00979-f004:** Impact of PGC-1α Knockdown on GBM Cell Viability. Viability following PGC-1α siRNA transfection in GBM cells (*n* = 3). Cell viability is shown for the control, negative control, and si-PGC-1α-transfected groups for GBM8401 and A172 cells at 1, 2, and 3 days post-transfection. The results indicate a significant decrease in cell viability in the si-PGC-1α groups over time (* *p* < 0.05, ** *p* < 0.01, and *** *p* < 0.001).

**Figure 5 biomedicines-12-00979-f005:** Migration Assay of GBM Cells Following PGC-1α Knockdown. A wound healing assay post-PGC-1α siRNA transfection in GBM cells (*n* = 6). The assay was conducted on GBM8401 and A172 cells divided into three groups: control, negative control, and si-PGC-1α. Time-lapse images were captured at 0, 8, 12, and 24 h post-transfection to assess the migration rate. Bar graphs quantify the percentage of wound closure, with significant inhibitory effects observed in the si-PGC-1α group compared with estimates in controls at later time points. Statistical significance was observed at 12 and 24 h post-transfection (* *p* < 0.05, ** *p* < 0.01, *** *p* < 0.001).

**Figure 6 biomedicines-12-00979-f006:** Invasion Assay of GBM Cells Following PGC-1α Knockdown. Transwell invasion assay following PGC-1α siRNA transfection in GBM cells (*n* = 4). Displayed are images of GBM8401 and A172 cells from the control group, negative control group, and si-PGC-1alpha group. The bar graphs to the right show the percentage of invasion, revealing a significant reduction in cell invasion in the si-PGC-1α group compared with the control group in GBM8401 and A172 cells (*** *p* < 0.001).

**Figure 7 biomedicines-12-00979-f007:** Protein Expression Analysis of N-cadherin, E-cadherin, Cyclin D1, and VEGF Following PGC-1α Silencing in GBM Cells. Western blot analysis of N-cadherin, E-cadherin, Cyclin D1, and VEGF following PGC-1α siRNA transfection in GBM cells (*n* = 3). Protein expression levels were assessed in GBM8401 and A172 cells for the control, negative control, and si-PGC-1α-transfected groups. The bar graphs below the blots quantify the relative expression normalized to β-actin. * *p* < 0.05, ** *p* < 0.01 and *** *p* < 0.001 compared with the control group.

**Figure 8 biomedicines-12-00979-f008:** In Vivo Tumor Growth Following PGC-1alpha Knockdown. Effects of PGC-1α knockdown on tumor progression in GBM cells in vivo (*n* = 6). After intracranial implantation of GBM8401 cells with fluorescence in the striatum, (**A**) the comparative epi-fluorescence intensity was evaluated. (**B**) progression over time post-implantation in the PGC-1α knockdown and control groups. *** *p* < 0.001 compared with the control group on Day 14 and Day 21.

**Table 1 biomedicines-12-00979-t001:** Correlation of PGC-1α expression with clinicopathologic parameters in patients with glioma.

	No. of Patients	PGC-1α Expression (*n*, %)	*p*-Value
		Low	High	
Age				0.342
<60	71	29 (30.5%)	42 (44.2%)	
≥60	24	7 (7.4%)	17 (17.9%)	
Gender				1.000
Male	52	20 (21.1%)	32 (33.7%)	
Female	43	16 (16.8%)	27 (28.4%)	
WHO Grade				0.003 **
II	23	15 (15.8%)	8 (8.4%)	
III/IV	72	21 (22.1%)	51 (53.7%)	
Tumor size				1
≤3 cm	59	22 (23.2%)	37 (38.9%)	
>3 cm	36	14 (14.7%)	22 (23.2%)	
Radiotherapy				1.000
No	54	20 (21.1%)	34 (38.9%)	
Yes	41	16 (16.8%)	25 (26.3%)	
Chemotherapy				0.052
No	59	27 (28.4%)	32 (33.7%)	
Yes	36	9 (9.5%)	27 (28.4%)	
KPS				0.643
≤70	67	24 (25.3%)	43 (45.3%)	
>70	28	12 (12.6%)	16 (16.8%)	

** Statistical significance (*p* < 0.01). WHO, World Health Organization; KPS, Karnofsky performance score.

**Table 2 biomedicines-12-00979-t002:** Univariate and multivariate Cox regression analyses of prognostic parameters in patients with glioma.

	Univariate Analysis		Multivariate Analysis
	Relative Risk	95% CI	*p*	Relative Risk	95% CI	*p*
Age	0.648	0.311–1.351	0.247			
Gender	0.892	0.473–1.682	0.724			
WHO grade	0.382	0.171–0.851	0.019 *	0.546	0.223–1.335	0.185
Tumor size	1.114	0.572–2.171	0.751			
Radiotherapy	1.845	0.965–3.526	0.064			
Chemotherapy	1.542	0.776–3.065	0.217			
KPS	1.167	0.586–2.324	0.660			
PGC-1α expression	0.177	0.075–0.417	<0.001 ***	0.197	0.080–0.482	<0.001 ***

* Statistical significance (*p* < 0.05). *** Statistical significance (*p* < 0.001). CI, confidence interval; WHO, World Health Organization; KPS, Karnofsky performance score.

## Data Availability

Data are contained within the article.

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
