# Peer review of "High PGC-1α Expression as a Poor Prognostic Indicator in Intracranial Glioma"

_biomedicines, 2024, doi:10.3390/biomedicines12050979_

Round 1

Reviewer 1 Report

Comments and Suggestions for Authors

Here, the authors discuss the role of PGC-1α in glioma progression, revealing that high expression of PGC-1α is associated with a worse prognosis and more advanced tumor stages. The researchers used various methods such as immunohistochemistry, cell culture, and siRNA transfection combined with functional assays to investigate the functions of PGC-1α in glioma cells. They propose that targeting PGC-1α could be a strategy to hinder glioma progression and enhance patient survival rates. Additionally, PGC-1α is suggested as an independent biomarker for glioma prognosis, offering potential for new treatments for glioblastoma.

Major points:

The authors provide a detailed examination of PGC-1α’s role in glioma, using a variety of methods, such as patient's data, in vitro and in vivo models. 

Their findings could have significant implications for glioma treatment and patient prognosis.

Minor points:

N numbers for each of the experiments are missing in the figure legends, and would help if present there. Scale bars are also missing in some figures (Figure 1/ 5/ 6). 

The authors should discuss the limitations of the study, and propose potential future steps, as more studies are required to validate PGC-1α as a therapeutic target and to explore its role in other cancers.

Comments on the Quality of English Language

minor typos

Author Response

Major points:

The authors provide a detailed examination of PGC-1α’s role in glioma, using a variety of methods, such as patient's data, in vitro and in vivo models. 

Their findings could have significant implications for glioma treatment and patient prognosis.

Thank you for your thoughtful summary and positive feedback on our work. We appreciate your recognition of the comprehensive methods we employed and the potential implications of our findings for glioma treatment and prognosis.

Minor points:

N numbers for each of the experiments are missing in the figure legends, and would help if present there. Scale bars are also missing in some figures (Figure 1/ 5/ 6). 

Thank you for pointing out the missing details in our manuscript. We have now added the N numbers for each experiment to the figure legends. Additionally, we have included scale bars in Figures 1, 5, and 6 as suggested.

The authors should discuss the limitations of the study, and propose potential future steps, as more studies are required to validate PGC-1α as a therapeutic target and to explore its role in other cancers.

Thank you for your valuable comments on our manuscript. We acknowledge the importance of discussing the limitations of our current study. Indeed, the limitations we recognize is that our study did not extensively explore the mitochondrial signaling pathways involved in the regulatory effects of PGC-1α on GBM progression. Moving forward, we aim to address this limitation by incorporating specific inhibitors and utilizing bioenergetic assessment tools, such as the Seahorse Bioanalyzer, to precisely delineate the pathways through which PGC-1α influences tumor progression in GBM. These studies will allow us to understand the intricate mitochondrial mechanisms influenced by PGC-1α and to further validate its role as a therapeutic target. Additionally, based on the results we obtain, we plan to expand our research to explore the role of PGC-1α in other types of cancer, providing a comprehensive understanding of its broader implications in cancer metabolism. We appreciate your suggestion and will include a detailed discussion of these future steps in our revised manuscript.

Reviewer 2 Report

Comments and Suggestions for Authors

Hello,

1.     In table 1 and 2, the p-value of couple parameters are not significant. How will you justify them?

2.     For your migration experiment, why did you choose 8hr time-point instead you could have used 6hr to show the liner time intervals?

3.     Can you please provide the images containing the marker with your respective Western blot bands.

4.     Have you repeated all the experiments at least three times? I can see only one experiment blots, if possible, kindly provide the repeated experiments blots as well.

5.     Discussion part: kindly elaborate your discussion part.

A.    In first paragraph, you have talked so much about glycolysis, metabolic activity, and Warburg effect; however, none of those you have studied in your study. Instead, you have shown clinicopathological characteristics and prognostic parameters, so kindly include refences related to them.

B.    Similarly, kindly incorporate more studies related to your rest of the animal work and other carcinogenic markers.

Comments on the Quality of English Language

Kindly polish your English language. Also, there are some words you must choose accordingly at sentence no 44 “necessary to for biomarker” and sentence no 109 “Fo transfection”, it should be ‘For’.

Author Response

  1. In table 1 and 2, the p-value of couple parameters are not significant. How will you justify them?

Thank you for your observation regarding the non-significant p-values for certain parameters in Tables 1 and 2. The study was primarily designed to investigate novel biomarkers and pathways with our sample size calibrated to detect major effects pertinent to the primary outcomes. Although the sample size was adequate for these primary measures, it might have been less so for capturing more subtle effects in all parameters. This acknowledged limitation is one aspect that we plan to address in future work by increasing the cohort size to enhance the detection of these smaller effects. Additionally, it is worth noting that the non-significant findings from the statistical analysis do not necessarily detract from the potential biological relevance of these parameters. Our study's strength lies in the supportive in vitro and in vivo experimental evidence that reinforces the role of PGC-1α in GBM progression. This combination of approaches provides a robust framework for our conclusions and suggests avenues for further investigation, including those parameters where statistical significance was not achieved. Understanding that the complexity of biological systems often yields data that may not strictly adhere to statistical expectations, we remain confident in the validity of our overarching hypothesis and findings. We will ensure to include a more detailed discussion of this context in the manuscript to provide a clearer picture of how these parameters fit within the larger scope of our research.

  1. For your migration experiment, why did you choose 8hr time-point instead you could have used 6hr to show the liner time intervals?

Thank you for your inquiry regarding the time-point selection for our migration experiment. We chose the 8-hour mark after careful consideration of our preliminary data, which indicated that this interval allowed for clear discrimination between different levels of cell migration activity. Although a 6-hour time-point would have provided a linear sequence in time intervals, our pilot studies suggested that the differences in cell migration were not as pronounced at this earlier stage. By extending the duration to 8 hours, we were able to observe a more robust migratory response, which we believe provides a more accurate reflection of the cellular behaviors we intended to measure.

  1. Can you please provide the images containing the marker with your respective Western blot bands.

Thank you for your request for the images of the markers alongside the Western blot bands. I would like to clarify that our current chemiluminescence detection system for Western blots does not have the capability to simultaneously display both the markers and the bands in a single image due to its design limitations. However, we have ensured that the molecular weights of the bands were carefully determined by comparing their migration distances to those of the markers run on the same gel.

  1. Have you repeated all the experiments at least three times? I can see only one experiment blots, if possible, kindly provide the repeated experiments blots as well.

Thank you for your inquiry regarding the replication of our experiments. Yes, all experiments were conducted in triplicate to ensure reproducibility and reliability of the data. I have now attached the additional Western blot images from the repeated experiments for your review.

  1. Discussion part: kindly elaborate your discussion part.

  1. In first paragraph, you have talked so much about glycolysis, metabolic activity, and Warburg effect; however, none of those you have studied in your study. Instead, you have shown clinicopathological characteristics and prognostic parameters, so kindly include refences related to them.
  2. Similarly, kindly incorporate more studies related to your rest of the animal work and other carcinogenic markers.

Thank you for your insightful comments regarding the inclusion of glycolysis, metabolic activity, and the Warburg effect in the discussion of our manuscript. You are correct in noting that our current study's primary focus was on the clinicopathological characteristics and prognostic parameters associated with PGC-1α expression in cancer, particularly in the context of GBM. This discussion serves to situate our findings within the broader context of PGC-1α research, drawing connections between our clinical observations and the underlying metabolic processes that are well-documented in the literature. We will revise the manuscript to include additional citations that are specific to the clinical aspects of PGC-1α. Furthermore, we plan to address the mechanistic aspects of PGC-1α’s involvement in glycolysis, metabolic activity, and the Warburg effect in future studies. These are indeed part of our research roadmap, as we aim to dissect the multifaceted roles of PGC-1α in the metabolic reprogramming of cancer cells. Understanding these processes will undoubtedly enrich our knowledge of PGC-1α as a biomarker and its implications for cancer therapy. In addition, I added “PGC-1α emerges as a pivotal figure in cancer biology, influencing a spectrum of me-ta-bolic processes and playing contrasting roles in tumor progression and response to therapy. In melanoma, its expression level bifurcates cells into two distinct metabolic states, with high PGC-1α levels fostering oxidative metabolism and resistance to stress, whereas low levels render cells glycolytic, prone to apoptosis, yet more invasive [43-47]. Breast cancer cells leverage PGC-1α's regulatory interactions to enhance mitochondri-al function and drive growth, with implications for aggressiveness and resistance to treament [48-49]. Co-activation between PGC-1α and c-MYC balance steers cell fate between survival and death, profoundly affecting treatment responses in pancreatic adenocarcinoma [50]. Similarly, in prostate cancer, PGC-1α dictates divergent paths of tumor behavior, with its levels modulating cell proliferation and invasiveness [51-53].” in manuscript.

Kindly polish your English language. Also, there are some words you must choose accordingly at sentence no 44 “necessary to for biomarker” and sentence no 109 “Fo transfection”, it should be ‘For’.

Thank you for bringing these oversights to our attention. We have carefully reviewed the manuscript and have corrected the language issues, including the typographical errors in sentences 44 and 109. We have taken additional steps to ensure that the language throughout the document is polished and that it meets the high standards expected of scientific communication.
